# Tools for Edible Cities: A Review of Tools for Planning and Assessing Edible Nature-Based Solutions

Eric Mino [1],*, Josep Pueyo-Ros [2,3], Mateja Škerjanec [4], Joana A. C. Castellar [2,3], André Viljoen [5], Darja Istenič [4], Nataša Atanasova [4], Katrin Bohn [5] and Joaquim Comas [2,3,6]

1 SEMIDE/EMWIS, Technical Unit of the Euro-Mediterranean Water Information System, BP23- Place Sophie Laffitte, Sophia Antipolis, 06560 Valbonne, France

2 Catalan Institute for Water Research (ICRA), Carrer Emili Grahit 101, 17003 Girona, Spain; jpueyo@icra.cat (J.P.-R.); jcastellar@icra.cat (J.A.C.C.); jcomas@icra.cat (J.C.)

3 University of Girona, Plaça de Sant Domènec 3, 17004 Girona, Spain

4 Faculty of Civil and Geodetic Engineering, University of Ljubljana, Jamova 2, 1000 Ljubljana, Slovenia; mateja.skerjanec@fgg.uni-lj.si (M.Š.); darja.istenic@fgg.uni-lj.si (D.I.); natasa.atanasova@fgg.uni-lj.si (N.A.)

5 School of Architecture, Technology and Engineering, University of Brighton, Lewes Road, Mithras House, Brighton BN2 4AT, UK; a.viljoen@brighton.ac.uk (A.V.); k.bohn@brighton.ac.uk (K.B.)

6 LEQUIA, Institute of the Environment, Universitat de Girona, C/Maria Aurèlia Capmany 69, 17003 Girona, Spain

* Correspondence: e.mino@semide.org

**Abstract:** In the last five years, European research and innovation programmes have prioritised the development of online catalogues and tools (handbooks, models, etc.) to facilitate the implementation and monitoring of Nature-Based Solutions (NBS). However, only a few catalogues and toolkits within European programmes are directly related to mainstreaming of NBS for food production (i.e., edible NBS). Therefore, the main aim of this paper is to present existing NBS tools through the eyes of productive urban landscapes. We reviewed 32 projects related to NBS and 50 tools were identified and characterised. Then, the six tools already available and provided indicators were further analysed in terms of their format and knowledge domains. Our main conclusion demonstrates that there is a lack of tools capable of supporting users for planning and implementing edible NBS; calculating the food potential of a city and/or of individual edible NBS, including the needed resources for implementation and operation (water, nutrients, energy); and assessing their urban design value, environmental and socio-economic impacts. Moreover, when they do exist, there is a resistance to share the models and equations behind the tools to allow other projects to reuse or validate them, a fact which is contrary to the open science principles upheld by many public research agencies.

**Keywords:** nature-based solutions; productive urban landscapes; decision support systems; edible cities; urban agriculture; circular economy

## 1. Introduction

By 2050, 68% of the global population is projected to live in cities [1], highlighting the relevance of urban ecosystem services to support not only the quality of liveable urban spaces, but also local food provision and resilient urban food systems [2]. Therefore, several "environmentally friendly" concepts have emerged to address the side-effects of high urban population densities, especially regarding increased resource consumption and environmental degradation. Consequently, cities are starting to evolve their infrastructures to work in line with the preservation of and increase in urban natural capital. At the urban planning scale, key examples of such concepts are green infrastructure [3–5] and multifunctional landscapes [6,7]. These concepts have been put forward by different disciplines, such as planning, landscape architecture, ecology, biology, forestry, and transportation, and due to their novelty, their definitions are still evolving [8]. However, neither of these green

concepts explicitly mention urban food production or other aspects of the urban food system [9].

In parallel with the above, several urban design approaches have emerged to articulate how urban food systems, and especially food production, can contribute to green infrastructure or multifunctional landscapes, e.g., *Continuous Productive Urban Landscapes* [10], *Sitopia* [11], *Agrarian Urbanism* [12] and *Food Urbanism* [13]. These concepts and approaches include diverse urban agriculture typologies and food system activities as widely documented, for example, in [14–18]. Spatial typologies, also known as edible green infrastructure [17], include allotments, urban farms, community gardens, school gardens, domestic gardens, edible urban forests, rooftop farms and vegetable raingardens, edible green walls and façades [17], whereas growing typologies include hydroponic and aquaponic systems, open and covered rooftop farms and soil-based food growing. As an urban design typology, these interventions are referred to as productive urban landscapes.

Water is a common driver for all these concepts. Thus, sustainable water management is of key importance for these concepts to function; therefore, a paradigm shift for resources management is needed in all related disciplines. Green infrastructure should transform into blue-green and urban agriculture should be seen as part of water and nutrient management in cities.

All these concepts and approaches also qualify as nature-based solutions (NBS), which conceptualise technological/spatial units and interventions [8], landscapes and actions as *'solutions inspired and supported by nature'* [19]. To highlight the necessity of engaging with the urban food system and to better distinguish different NBS, we propose the term *'edible NBS'* when referring to NBS that have the purpose of food production. Similar to NBS, edible NBS have great potential to provide a series of co-benefits (e.g., wellbeing and biodiversity), enhancing the natural capital and management of urban resources but with a focus on the food system. This mainly concerns water, nutrients, and energy and can contribute to a more circular urban metabolism and circular economy principles [20].

In the last five years, European research and innovation programmes have prioritised the development of online catalogues and tools (handbooks, models, etc.) for NBS, mainly due to the demand of public, private and research organisations concerned with the existing gap of knowledge and successful case studies regarding the planning, implementation, and monitoring of NBS for the development of more sustainable and resilient cities. The tools are now so numerous that even a catalogue of tools was developed to assist users in selecting the most appropriate one for their needs. This catalogue, developed in the project ACTonNBS (Adaptive Cities Through Nature Based Solutions), includes around 70 tools to take up NBS and enhance climate resilience in cities. Even though urban food production is beginning to be recognised as an important component of sustainable urban ecosystems, most of the current available tools on NBS support the design, implementation and monitoring of NBS in general, with no special focus on the edible potential of a number of these solutions. Only a few catalogues and tools within European programmes are directly related to mainstreaming edible NBS. Such resources are still mainly found in connection with individual projects by urban agriculture practitioners, such as Farming Concrete in New York (https://farmingconcrete.org/, accessed on 25 August 2021), and/or are linked to individual cities. Moreover, there are several indicator frameworks to assess urban water management (e.g., the City Blueprints) or city sustainability (e.g., Green City), where NBS impacts are directly included (e.g., via percentage of green spaces). However, edible facets are not explicitly included in these frameworks, although they would have a direct impact on some of their indicators.

Therefore, the main aim of this paper is to present existing NBS/edible NBS tools through the eyes of productive urban landscapes. To the best of our knowledge, so far, no publication reviews the existing tools to support the implementation of edible NBS as well as its features, content, and facets. In this sense, we reviewed 50 identified tools in 32 different research and development projects financed by national or international organisations such as Non-Governmental Organizations (NGO), governmental organisa-

tions, and the European Commission, with three main purposes: (1) to identify and classify operational tools with potential to support edible NBS implementation and impacts; (2) to know to what extent those tools could be (maybe partially) adapted or reused by other projects; and (3) to identify the gaps in existing tools, and then, the need for developing new tools for edible NBS, with a holistic focus from planning to implementing, monitoring and assessment, taking into account resource management in cities. In particular, we were interested to what extent (if any) these tools provide information about resources use, i.e., water, nutrients, energy, and the connection between edible NBS and water management thereof. Additionally, we analysed which indicators have already been used for such purposes and selected a set of indicators that most holistically assess edible NBS. In addition, all underlying data are provided as Supplementary Material.

## 2. Conceptualisation

To facilitate a better understanding of our study, in this section, we briefly explain the terms Tools (Section 2.1) and Indicators (Section 2.2), both applied under the framework of planning, design, implementation and monitoring of edible NBS, as environmental measures to address urban challenges.

### 2.1. What Do We Mean by Tools?

In environmental sciences, there are different understandings and classifications of tools and no unique definition. Moreover, the complexity level, structure and purpose of the tools can vary greatly. For example, more complex tools are based on analytical methods (e.g., Life Cycle Analysis), indicators and/or mathematical models (in the form of mathematical equations) to design, plan and/or evaluate a specific socioeconomic and environmental performance (e.g., Parcels: https://parcel-app.org/, accessed on 22 December 2020). In contrast, less complex tools can be based on informative approaches such as web-based catalogues of information (e.g., NBS knowledge hub: https://platform.think-nature.eu/nbs-projects, accessed on 12 February 2021), which can offer search options (e.g., Naturvation Atlas: https://naturvation.eu/atlas, accessed on 12 February 2021; EdiCitNet Toolbox: https://toolbox.edicitnet.com, accessed on 12 February 2021). Nevertheless, all types of tools, regardless of their level of complexity, are usually designed clusters of information or software applications based on information and communication technologies that help in decision making, planning, design or evaluation of socio-economic and environmental impact.

In this paper, we divided the tools dealing with NBS into two major groups: (1) information-based tools that include different textual, visual and/or graphical data, and in most cases, also search options, and (2) model-based tools that include one or more mathematical models to calculate specific outputs. Therefore, a tool in this context provides users with information and/or quantitative assessments based on provided input data. These assessments are related to food production/activities, impacts (environmental, urban design, social, economic) and/or estimation of relevant indicators for sustainable resource management in cities.

### 2.2. What Do We Understand as Indicators?

According to the Organisation for Economic Co-operation and Development (OECD) [21], an indicator is a parameter, or a value derived from other parameters, which provides information about the state of a specific system. The impacts of a particular environmental measure (e.g., edible NBS) can be assessed quantitatively and/or qualitatively by adopting key performance indicators (KPIs), a set of parameters providing the means to assess particular attributes to meet an explicit objective. In this regard, the performance of an edible NBS can be defined as the degree to which edible NBS addresses urban challenges (e.g., climate resilience or social justice) and/or fulfils a specified objective in a specific context. The KPIs can be, for example, biophysical, social, or economic indicators, which are targeted for specific aspects of edible NBS's effectiveness. In addition, a new generation

of indicators is emerging that are also looking at institutional and governance aspects [22]. Above all, they reflect the need for cities to transition from linear to circular management of resources, encompassing a cross-sectoral and cross-disciplinary approach. Nika et al. [23] propose a circular economy indicator framework for complex water systems incorporating metrics for interrelation between water and other relevant urban sectors, including urban agriculture.

If several KPIs are applied, we refer to indicator frameworks (also termed indicator sets, or systems), which can be used for monitoring and providing the feedback needed to accomplish the desirable state of urban sustainability [24]. Moreover, indicator frameworks can be used to evaluate the impact of specific measures at a city scale. Some examples are City Blueprints [25], EEA Urban Metabolism Framework [26], European Green City Index [27], and the newly proposed Circularity Assessment Framework for complex Water Systems by Nika et al. [23], which are able to compare performance between similar cities [28].

## 3. Materials and Methods

The reviewed projects were selected using the NBS projects database elaborated under the scope of Cost Action Circular Cities (CA17133) [29], which was extended to other projects related specifically to edible NBS identified by the partners of the EdiCitNet project (EU H2020 project GA 776665). Two criteria were used for this selection: (1) projects offering tools focused on urban farming; and (2) projects offering tools focusing on NBS but potentially enabling urban farming. The tools provided or used by each project were identified and analysed by browsing project websites and reading related publications. A new database was created with analysed projects and related tools. This database is provided in the Supplementary Material.

In the following sections, we describe the characterisation of the identified (information- and model-based) tools according to five main characteristics (Section 3.1); the systematic selection of tools based on their accessibility, operability and the availability of data and calculation methods (Section 3.2); and description of the parameters used for further analysis of the finally selected (model-based) tools (Section 3.3).

### 3.1. Characterisation of Tools

To characterise different tools, we applied an end-user-centred approach, by imagining the scenarios under which users with different backgrounds and aims (e.g., urban planners, scientists, civic society, community-led organisations) might wish to access the content and format, most appropriate to their needs. For example, an agronomist seeking specific crop yield data would require production metrics (e.g., quantitative data), whereas a community-led organisation may find a narrative description about the production more useful. In this regard, we departed from Katsou et al. [29] and formulated five main characteristics of the identified tools:

1. Typology of the tool: Information-based tools, which organise and display information by providing visual and consultative outputs such as catalogues or handbooks, or model-based tools, which provide quantitative estimations of performance and impacts expressed as indicators, models or equations (e.g., estimating yield or water needs).
2. Geographical level of policy or regulation: Local or regional, national, European or international.
3. Phases of edible NBS's full cycle implementation assessed by the tool: Planning and Design (e.g., estimating needed resources or aiding design), operation and monitoring (e.g., data collection, operational and maintenance tasks such as harvesting or events), assessment (impact, performance indicators) and communication (e.g., aiding in the dissemination of edible NBS).
4. Sustainability dimensions addressed: Social (addressing aspects related to wellbeing and equity, social cohesion, cultural values), economic (dealing with aspects related

to job creation or businesses' potential) or environmental (addressing environmental aspects such as carbon sequestration, air quality, water management or biodiversity).

5.   Type of provided support or stakeholders' engagement: Inform (one-way communication from project to citizens, e.g., handbook), consult (two-way communication where stakeholders can provide their opinions, e.g., survey), involve (stakeholders are passively engaged in the project, e.g., focus groups), collaborate (stakeholders are actively engaged in the project, e.g., collecting data) or train (the tool is used to enable skills and capabilities).

The tools were characterised using publicly available information, tool testing, when possible, and direct contact with tool developers (email exchanges and virtual meetings), when necessary.

### 3.2. Systematic Selection of Tools

From the tools identified in the reviewed projects, we selected a reduced number of tools according to the following criteria:

- Accessibility: the tools can be used by the general public (through open access or by using a license).
- Fully operational: the tools are finished and fully working.

In order to analyse to what extent the tools could be reused, the data were further analysed considering inputs (data provided by users or external databases used) and outputs (indicators or variables computed by the tool) as well as potential geographical constraints for reuse of the tool (e.g., use of external datasets existing only in some countries).

Then, for further analysis, we selected only the tools that were model-based in order to extract the indicators that those were providing. Therefore, the main outcome was one or several indicators calculated based on the input data. A graphical representation of the method applied to characterise and select the tools for further analysis can be seen in Figure 1.

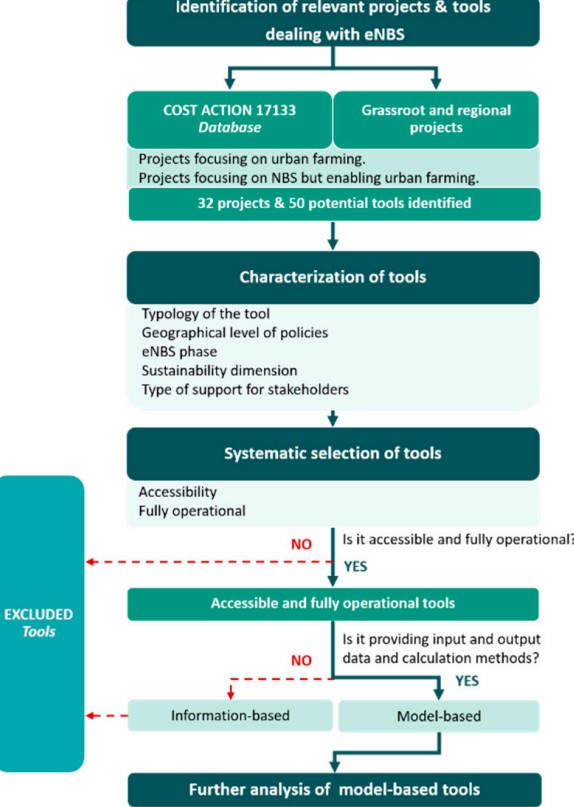

**Figure 1.** Graphical representation of the method applied to select, characterise and analyse the tools.

### 3.3. Further Analysis on Model-Based Tools

In undertaking this review and analysis, we were conscious of the different end users likely to use the tools, primarily from the environmental sciences regarding quantifying impacts or from the design disciplines regarding how edible NBS and productive urban landscapes can be implemented. The different cognitive approaches of scientific and design disciplines [30,31], whereby scientists often start from a problem analysis perspective, often quantifiable, and designers start from a solution-oriented approach, informed the way we analysed existing tools, seeking to evaluate them in relation to the kinds of content different users would find useful. This led to a matrix of categories generated through an iterative process of deep analysis from which we defined knowledge domains and format domains that address analytical approaches (e.g., metrics and systems) and solution approaches (e.g., design and policy). In doing this, we acknowledge the fuzzy nature of this categorisation, an objective being to encourage the breaking down of artificial disciplinary boundaries. The first axis of the matrix represented the type of knowledge sought (knowledge domains) and the second axis represented the format within which the knowledge was made available (format domains). This provided 8 categories for Knowledge Domains (Environment, Water, Social, Economic, Systems, Policy, Production and Spatial) and 9 categories for Format Domains (Typology, Design, Social Media, Guidance, Metrics, Narrative, Impact, Bottom-Up, Top-Down), resulting in a 63-cell table, each cell representing a particular knowledge type and format for that knowledge (Table 1).

**Table 1.** Description of matrix categories used to classify the analysed tools.

| | Domain | Topics |
|---|---|---|
| **KNOWLEDGE DOMAINS** | Environment | Tools covering the conceptual role of edible NBS in terms of enhancing environmental sustainability, e.g., preserving and enhancing biodiversity, promoting a sustainable drainage, or reducing diffuse pollution. |
| | Water | Tools dealing with water needs and/or water management in edible NBS. |
| | Social | Tools that relate to demographic aspects, groups or personal relationships and number of jobs created for a certain type of edible NBS. |
| | Economic | Tools dealing with economic values of edible NBS and economic mechanisms (nonprofitable, business-focused). |
| | Systems | Tools providing insights about interrelations between urban metabolism, circular systems and edible NBS. |
| | Policy | Tools providing examples and guidance in terms of urban planning and wellbeing policies related to implementation and functioning of edible NBS. |
| | Production | Tools providing information on production and processing of edible goods in edible NBS. |
| | Spatial | Tools dealing with information regarding size, arrangements, sites and locations of edible NBS. |
| **FORMAT DOMAINS** | Typology | Tools proposing a classification for edible NBS in terms of users' motivations, used technologies, urban design approaches, etc. |
| | Design | Tools showcasing case studies of landscape, architectural and urban design processes of edible NBS. |
| | Social media | Tools including an informal space for user interaction and knowledge exchange, such as blogs or forums. |
| | Guidance | Tools providing step by step guides in terms of, for example, gaining finance or improving yields. |
| | Metrics | Tools providing numeric evidence (raw data or indicators) in terms of, for example, biodiversity, number of users or yield. |
| | Narrative | Tools providing a narrative evidence of edible NBS (e.g., factsheets, case studies) in terms of, for example, biodiversity, number of users or yield. |
| | Impact | Tools providing an impact assessment (qualitative or quantitative) of edible NBS in terms of environmental and socio-economic effects and benefits. |
| | Bottom-up | Tools that include processes led by individuals or community groups. |
| | Top-down | Tools that include processes led by institutions, municipalities, governments, etc. |

By the nature of the subject, this categorisation is somewhat fuzzy but convenient in providing an overview of the domains in which the tools are useful. It is to be understood in relation to a user wanting to find out more about edible NBS.

## 4. Results

The following sections present the characterisation of the 50 identified tools, describe the systematic selection of the tools and provide details on the further analysis carried out for the model-based tools.

### 4.1. Characterisation of Tools

Most of the 50 tools reviewed are aiming to support decision-making processes with interactive software applications or guidance documents either in an informative way—providing guidance toolkits or case studies, sometimes supported by interactive catalogues enabling geographical or keyword search (33 tools)—or via mathematical models (e.g., for calculation of indicators), falling under the model-based category (17 tools).

The information-based tools include a wide range of knowledge, e.g., data and indicators, which are usually provided in textual documents (reports or case studies), and thus, more difficult to exploit than when resulting from model-based tools.

Most tools consider the importance of policy and regulation for the deployment of edible NBS (84%) by providing qualitative recommendations (e.g., to improve participation and governance) or impact indicators relevant for policy (e.g., Urban Foodprint). The recommendations are mainly referring to the local or regional level (60%), indicating the importance of policies directly connected to the implementation of edible NBS (Figure 2).

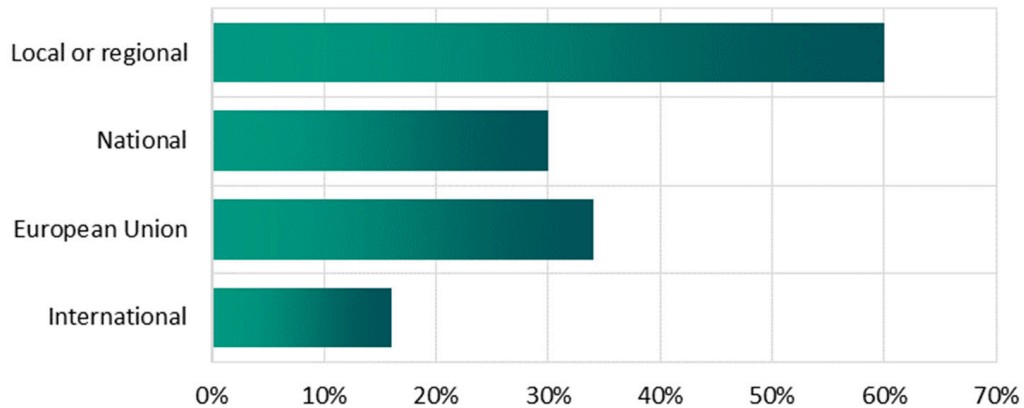

**Figure 2.** Level of policies addressed.

Sixty-two (62) percent of the tools deal with all three sustainability dimensions (Figure 3). However, the reviewed tools mostly address the environmental dimension of sustainability (86%). Moreover, even though social issues are the main concern of local governments when developing urban agriculture initiatives [32], this is less reflected in the support offered by the tools reviewed (74% vs. 86% on environmental issues). Likewise, although urban food production can improve the economic sustainability of a city [33], the economic dimension is least represented among the studied tools (66%).

The reviewed tools provide support for the different phases of edible NBS implementation, from planning and design, operation and monitoring to the assessment as well as communication (Figure 4). Very few are supporting all the phases (16%). In general, evaluated tools and, in particular, the ones with a social and economic dimension, are more oriented to planning/design and communication than to operation/monitoring and assessment.

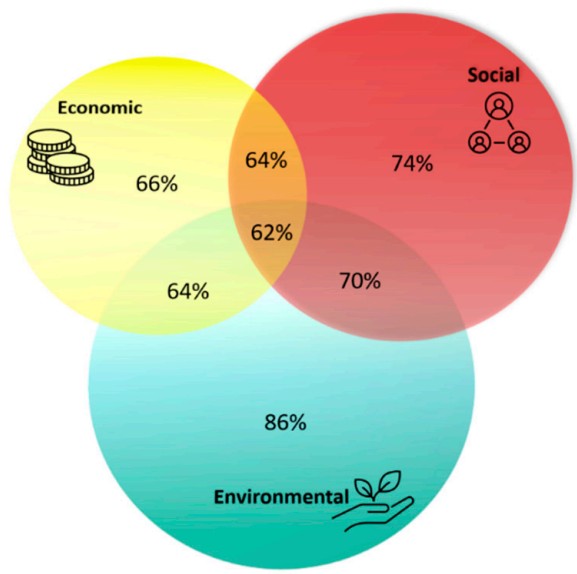

**Figure 3.** Sustainability dimensions covered by the tools.

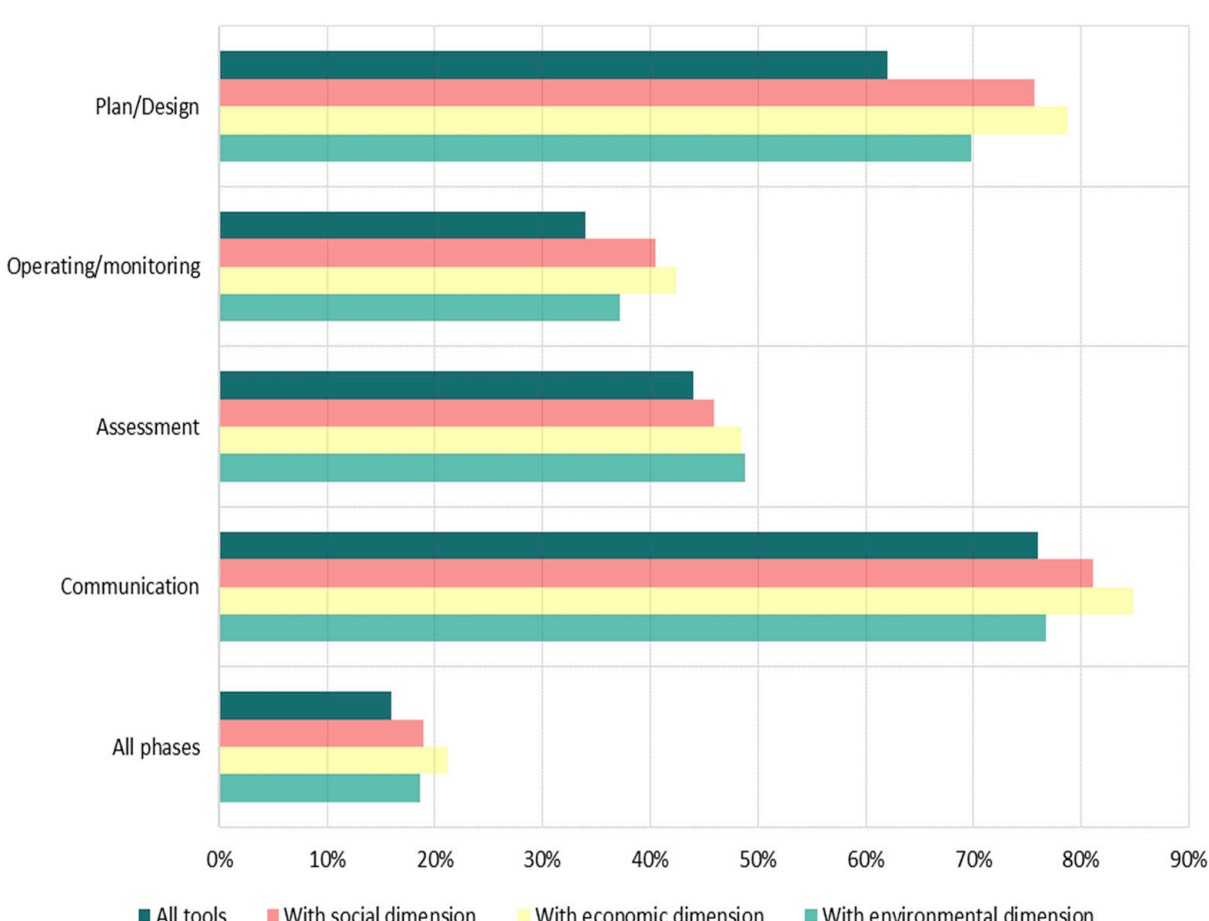

**Figure 4.** Phases of edible NBS implementation addressed by the tools.

Looking at the type of support the tools provide in terms of stakeholders' involvement (Figure 5), it appears that most of the tools support a one-way communication (information diffusion), while 13% of the tools provide no support for engagement. The majority of the tools addressing social issues provide support for community engagement. Tools with

social and economic dimensions (yellow and red bars in Figure 5) are more oriented to stakeholder involvement compared to tools with an emphasised environmental dimension (turquoise bars in Figure 5).

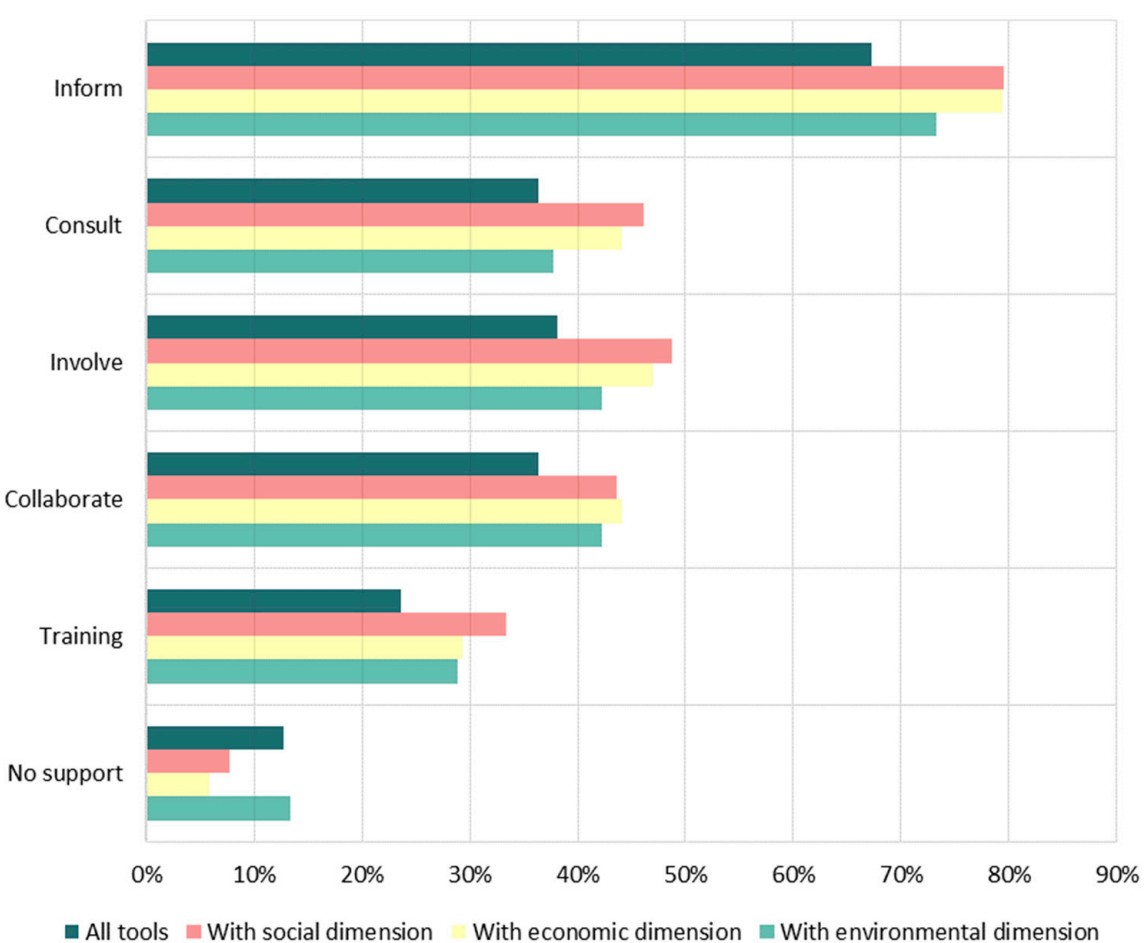

**Figure 5.** Stakeholder's level of involvement provided by the tools.

*4.2. Systematic Selection of Tools*

Out of 50 tools, 29 accessible and fully operational tools were shortlisted. A good indication of the rising importance of tools to support the implementation of edible NBS is that most of the model-based tools (66.6%) and information-based tools (69.5%) are focused on edible NBS (Figure 6), and not on general NBS that potentially can enable sustainable urban farming. Among these 29 tools, only 6 of them (approx. 20% of total tools) are model-based tools (Figure 6), mainly focusing on very specific components (e.g., crops or geothermal energy for food production or farm management) rather than providing a more holistic support to urban planners. The remaining 23 tools are information-based tools such as catalogues (case studies, projects, business models), guidelines, interactive forums, or networks (e.g., Red de Ciudades por la Agroecología). The level of interaction is variable, from classical bookshelves' consultation to multicriteria search (e.g., topics, location, type of stakeholder). This type of tool is very useful to gain knowledge (e.g., for defining indicators) and learn about know-how, but the lack of unified approach in the analytical method used may create confusion and difficulties in comparing the results obtained. In addition, searching for relevant information can be time consuming and can represent an important barrier for practitioners, in particular, when searching for food-related information within the NBS tools.

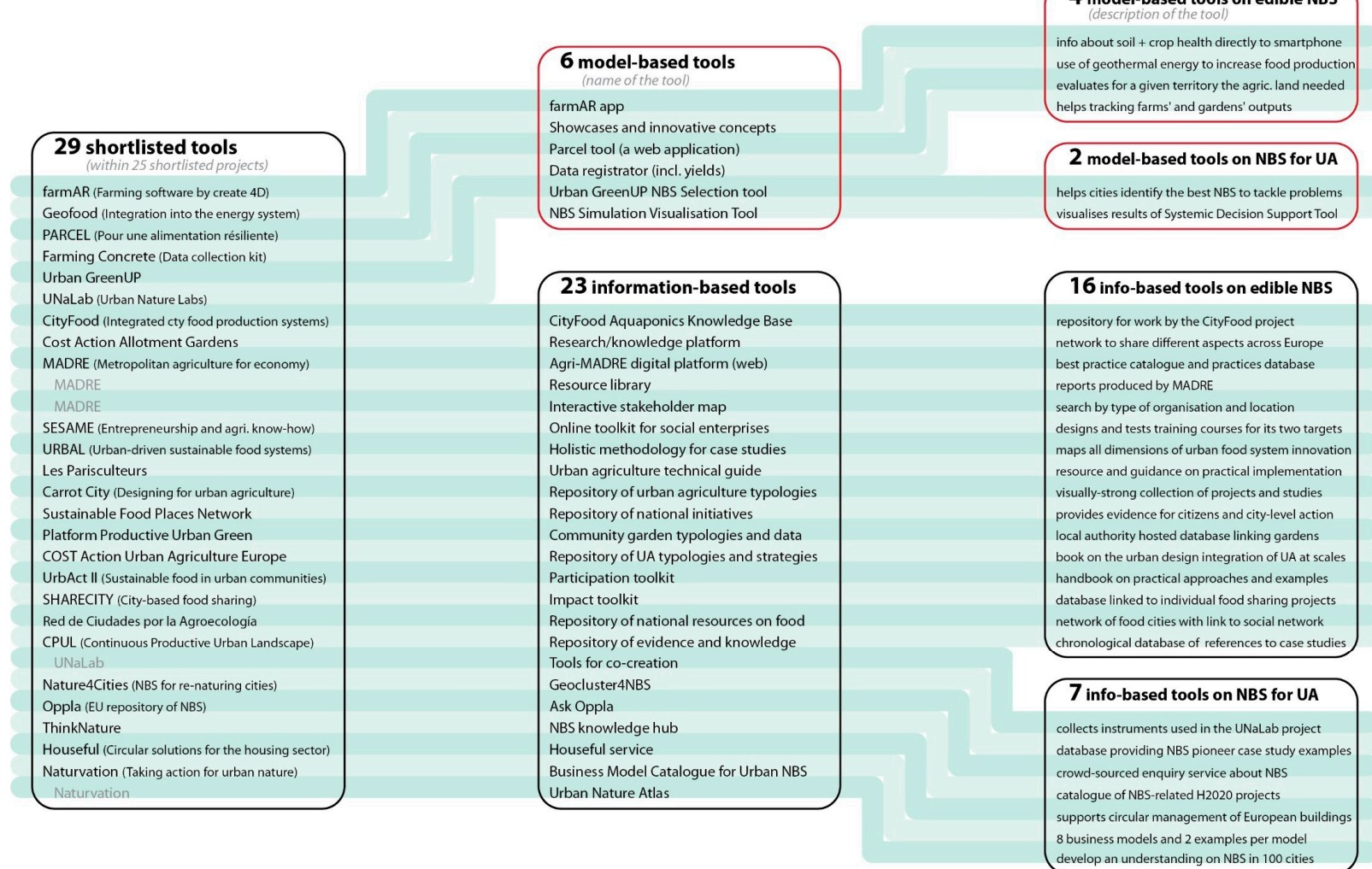

**Figure 6.** The 29 shortlisted tools supporting model-based and information-based decision processes for implementation of edible NBS and NBS enabling urban agriculture.

The screening of data and indicators used by the 29 accessible tools resulted in four main types of data (underlying data are provided in Supplementary Material:

- Input data provided by the end user to specify their needs, such as geographical location (e.g., city or region), boundaries of the area, (e.g., digital map layer), urban challenge to be addressed, and local population.
- Monitoring data: e.g., honeybee activity, "Lnight" and "Lden" for environmental noise (indicators defined by EU Environmental Noise Directive (END): for the day, evening and night periods (Lden) and for night periods (Lnight)), electrical conductivity, pH, or water infiltration rate.
- External data sources, such as meteorological data from public stations or satellite imagery.
- Calculated impact indicators, e.g., life cycle assessment/analysis, economic evaluation of the edible NBS, urban food production, urban food demand, space availability, carbon footprint, jobs created (full time equivalent), inflow of mass and energy, estimated limit of sustainable energy production and total energy in the reservoir, crop yield indicator, spatial footprint indicator, and cultivated area. The method used by the tools for calculating the impact indicators is usually not provided, except for grassroot tools such as Farming Concrete, allowing local communities to make rough estimates.

### 4.3. Further Analysis of Model-Based Tools

A short description of the six model-based tools selected for further analysis can be seen below:

- FarmAR app: A smartphone application that provides information about soil and crop health by using remote sensing product databases.
- Farming Concrete: A web-based toolkit calculating indicators based on data provided by more than 400 community gardens. Moreover, the toolkit offers a total of 18 very practical data collection methods to support local communities in monitoring their edible NBS. Collected data are organised into five categories: food production data, environmental data, social data, health data and economic data.
- GeoFood: This tool provides innovative concepts illustrating how to increase the economic viability of joining geothermal heat infrastructure and circular food production systems for aquaculture.
- UrbanGreenUP: NBS Selection tool (UGU NBS) recommends NBS for a selected city, based on specified challenges and the capabilities of a particular organisation. It helps cities to select the most appropriate NBS to tackle identified environmental problems and to become more resilient to climate change.
- UNaLab: NBS Simulation and Visualization Tool (UNL—NBS-SVT) can be used to visualise the results of the UNaLab systemic decision support tool for assessing the multiple impacts of NBS. Different scenarios can be simulated and visualised, i.e., a reference scenario, nature-based scenarios, population growth scenarios, climate change scenarios and combined scenarios. Additionally, results can be provided for different time periods.
- Parcels: For the selected area and number of inhabitants, this tool calculates the agricultural land needed to achieve food self-sufficiency. Moreover, it provides information on the agricultural jobs potentially created and the ecological impacts (e.g., greenhouse gas emissions, pollution of water resources, and effects on biodiversity) associated with possible changes in food production methods and/or in dietary habits.

The six model-based tools are fully described in Supplementary Material, with the list of data they are using and providing. Four tools (Parcels –"Pour une Alimentation Résiliente et Citoyenne et Locale", Fondation Terre de Liens, Fédération Nationale de l'Agriculture Biologique, Bureau d'Analyse Sociétale pour une Information Citoyenn, France-, UNaLab—Urban Nature Labs, NBS Simulation Visualization Tool, Engineering, Roma, Italy, University of Aveiro, Portugal-, UrbanGreenUp—NBS selection tool, Royal Melbourne Institute of Technology, Autsralia- and GeoFood—"Predictive models to design

thermal treatment for circular food production systems", University of Iceland, Reykjavik, Iceland) are ex ante tools supporting planning and design by, for example, allowing estimations of the requirements for moving towards food self-sufficiency (Parcels' tool), while two tools (Farming Concrete and FarmAR app) can support local practitioners in the daily management of their edible NBS by, for example, providing low-tech methods for collecting data and an online repository to store and aggregate crop yield data (Farming Concrete).

Looking at the metrics provided, the selected tools mainly focus on the spatial (e.g., area cultivated, 15 coloured target dots in Figure 7), production (e.g., yield, 12 dots in Figure 7) and environment (13 dots, e.g., water abstraction, 12 dots) knowledge domains, rather than on the social (4 dots), policy (10 dots) and economic (only 1 dot) knowledge domains (Figure 7). In contrast, the three tools falling under the narrative format domain (i.e., UNaLab, UrbanGreenUp and Parcels) are mainly devoted to policy topics. The Parcels tool covers the largest amount of knowledge domains in different formats, while other tools are narrower in their delivery of knowledge items (e.g., UNaLab). On the contrary, the GeoFood tool has a narrow knowledge focus, supporting the design phase of aquaculture. For a specific knowledge domain, the tools follow either a bottom-up or top-down approach, except for the spatial dimension where both approaches are necessary.

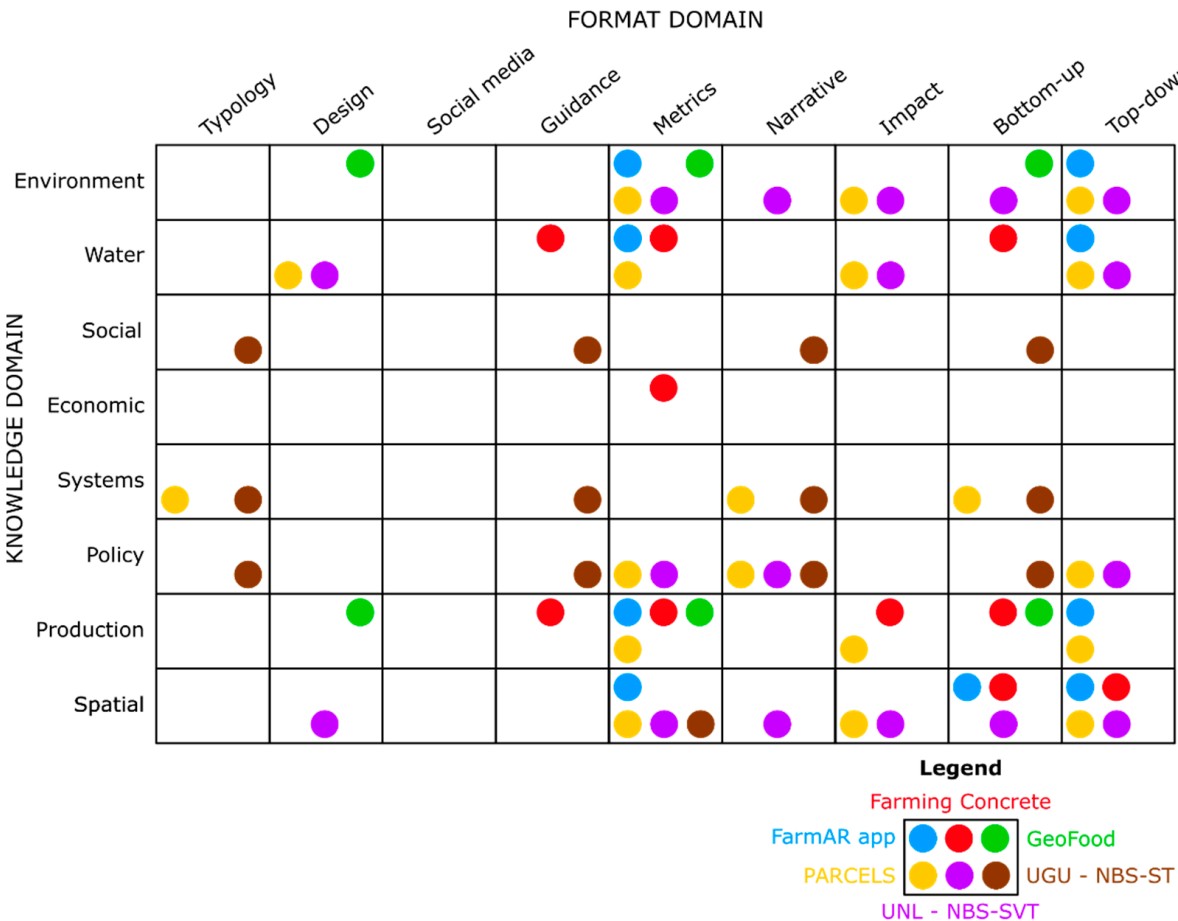

**Figure 7.** Location of model-based tools in a two-dimensional grid, where the *x* axis represents the format domains and the *y* axis represents the knowledge domains. Coloured dots represent the six different model-based tools further analysed.

Furthermore, the matrix indicates that design is only highlighted in three of the model-based tools, reinforcing the existing gap between science and design disciplines [30,31], and thus, highlighting limitations regarding the much-needed multidisciplinary approach to successful implementation of edible NBS. The absence of tools utilising social media and

promoting model-based tools further evidences the more science-based discourse about edible NBS.

The matrix also highlights that four tools (out of the six) provide data or indicators related to water management: Farming concrete provides guidance on how to measure water needs and a data repository on rainfall and rainwater harvested; the UNaLab tool uses models to simulate the impact of NBS on flooding; Parcels evaluates the water footprint of an urban agriculture project; and FarmAR estimates soil moisture based on satellite data to advise on irrigation needs.

## 5. Discussion

In the ACTonNBS project, which reviewed around 70 NBS tools to enhance climate resilience in cities, the tools were classified according to their purpose, where 32% of addressed tools provide planning and design support for NBS implementation, while other tools are informative (29%), enable analysis (20%) or are inspirational (18%) to foster NBS implementation [34]. Although our classification is different, we can draw a comparison. Namely, among the identified edible NBS tools, there is a much higher share of tools supporting planning and design (more than 60%) and assessment (more than 40%) compared to NBS tools for climate resilience. This indicates that edible NBS tools may be more practically oriented and more specific than general NBS tools, thus offering more tangible results to the end user. Furthermore, we found that topics covered in planning/design tools and in modelling tools do not refer to the interdependence of planning/design and modelling of edible NBS. In particular, we found no direct relations between the implementation of edible NBS and urban water management, despite their clear relation in terms of the fact that edible NBS are, in many cases, part of rainwater harvesting and their clear positive impact on sustainable urban drainage. This reveals significant potential for future research and practice aimed at breaking down disciplinary barriers. New methodologies and activities need to be developed to translate the highly specialised knowledge between different disciplines to enable efficient collaboration between disciplines, leading to sustainable implementation of edible NBS and productive urban landscapes and to objectively assess their impact on urban resource management.

Moreover, existing tools provide limited guidance concerning resources needed for sustainable urban agriculture and their management in the sense of resources recovery. This can be considered as an important gap, despite the significant role of urban agriculture in increasing the self-reliance of the urban food system. Implementation of urban agriculture depends on a great variety and amount of resources such as water, energy and nutrients [35]. Therefore, if edible NBS are to be integrated as a solution to sustainable resources management in cities, then the tools need to integrate knowledge from the field of water, energy and wastewater management, such as hydrology (surface runoff, water balance), and nutrient balance modelling for estimating the reuse potential (e.g., from wastewater). While models for sustainable urban water management are being developed that take into account wastewater reuse and rainwater harvesting [36], the review within this research reveals that none of the edible NBS tools include such integration. This is somehow expected, as it requires a strictly cross-disciplinary approach and, as such, represents a gap towards creating complete urban resources management-oriented tools. Another potential reason for a limited integration of environmental disciplines in the framework of existing tools is that edible NBS are still not seen as serious mitigation measures for resources management but rather, as tools for addressing the self-reliance of urban food systems, social challenges and integration [37–39]. This yet again calls for quantitative tools that can realistically estimate the impact of edible NBS on sustainable resource management (e.g., reducing the ecological footprint of cities).

Furthermore, there is a gap in assessing what users' needs are regarding the implementation of edible NBS and, thus, what the tools should provide to them. However, this leads to another discussion. Should tools be limited to satisfy the users' needs? Or should they go beyond to provide knowledge and skills that users did not know they needed? For

instance, users could be interested in knowing how much water they will need to grow food, but perhaps the tool should go beyond this and provide knowledge on the use of alternative sources of water such as rainwater or reused wastewater and, thus, reinforce circularity. Furthermore, not only urban agriculture users, but also water professionals should be aware of the benefits of edible NBS to the urban water cycle.

In this sense, EU H2020 project EdiCitNet (Edible Cities Network—Integrating Edible City Solutions for social, resilient and sustainably productive cities) provides a series of tools (via an open data portal: https://toolbox.edicitnet.com/, accessed on 12 February 2021) to help city planners, urban agriculture communities, and individuals to plan and assess edible NBS benefits in their cities and to facilitate sharing of data, knowledge and experiences. One of the tools is focused on providing guidance for design and planning of edible NBS, and is, thus, putting forward valuable insights about resources needed and expected food potential. Moreover, the EdiCitNet toolbox will also facilitate a participative planning approach and training of edible NBS dealing with urban challenges. Setting the objective to meet the needs of such a wide range of potential stakeholders is important at this stage in the evolution of edible NBS because the practice, although growing, is by no means normative and multiple players are seeking to evaluate the case for their implementation. As a result, the level of pre-existing knowledge of a potential user of the toolbox will vary from someone wishing to gain an overview to others requiring detailed knowledge.

Another way to quantify the impact of edible NBS on urban resources management is via indicators. Cities have developed different indicator schemes to assess their sustainability (e.g., City Blueprints [25], EEA Urban Metabolism Framework [26] and European Green City Index [27]). Although they include many indicators related to the sustainable management of water, indicators for the success of resources recovery, green spaces, and other relevant urban measures related to sustainability, urban agriculture is not explicitly included in these schemes as one of the measures for resource management or for urban sustainability. In addition, the handbook for NBS monitoring recently published illustrates the lack of specific indicators related to urban food production [40]. This review shows that only six tools related to edible NBS are potentially linked to city indicators. Thus, there is a need for an update of existing city sustainability indicators and consequently, a link of future edible NBS related tools with such indicator schemes.

Another relevant point is the lack of tools estimating the socio-economic impact or benefits of edible NBS, probably because these impacts are also hardly quantified for general NBS. Thus, there is a need for tools and models to be able to quantify NBS co-benefits and to illustrate whether these models can be directly applied to edible NBS or whether they will require some previous adaptation. Our deep analysis on the type of support provided also reveals a need for tools helping training and collaboration among stakeholders when dealing with edible NBS.

Moreover, one of the important aspects to consider when selecting a tool or an indicator framework is data availability. Without proper data, based on monitoring, it is not possible to develop indicators or use specific tools. There is often little or no consideration of what data are readily available when the indicator framework or tool is proposed. City Blueprints is a classic example: despite planning the indicator set around publicly available data, they struggled to obtain the data required to complete the assessment of Rotterdam's water sustainability [25].

Nonetheless, when the proper tools exist, an important issue is the difficulty in obtaining specific equations and models to replicate, reuse or validate those tools. Open science is a growing movement, and the advantages of open science are more than demonstrated. For instance, it has been evidenced how open science can accelerate research [41] or how it increases the impact of publications [42]. Likewise, most public funding agencies are including open science principles and requirements in their funded research, such as the FAIR principles advocated by the European Commission, whose main goal is the reuse of valuable research objects [43]. However, our experience in this review of tools was quite

the opposite. Since our goal was avoiding reinventing the wheel, we looked for and even, in some cases, explicitly asked for the models and equations behind the tools that could be useful for us, but no equations or models were given in almost any case (only UNaLab, UrbanGreenUp and Parcels provide documents describing the models in detail). Unfortunately, this is coherent with other studies that found that nearly 80% of research datasets are not properly shared [43]. Notwithstanding, this is surprising considering that most of the reviewed projects were funded by public agencies that promote open science. So, why researchers are not sharing their models with other projects that could reuse them? In some cases, there is a common perception that open practices could present a risk to career advancement [42]. In others, the researchers believe that sharing may foster unwelcomed competition [43]. However, this results in public agencies (i.e., taxpayers) paying more than once for the same job carried out by different researchers. Consequently, the avoidance of time-wasting and, thus, of public money, would be a first return on investment in real open science requirements [43].

Furthermore, perhaps the greatest challenge in comparing tools is to evaluate the reliability, breadth and depth of data presented in a tool as well as their user friendliness. Reliability and level of uncertainty are particularly important for decision making when designing and assessing edible NBS; however, the quality of the tools and data provided was out of the scope of this paper, which focuses on clustering and analysing the type of services provided.

## 6. Conclusions

Currently, there are a reduced number of tools, methodologies, and indicator schemes capable of supporting users in the planning, design, monitoring and assessment of edible NBS. To facilitate the transition towards more edible and resilient cities, such tools should be able to calculate the food potential of a city and/or of individual edible NBS, include the needed resources for implementation and operation (water, nutrients, energy), estimate how edible NBS are related to other urban sectors, such as urban water management and energy sectors, and assess their environmental and socio-economic impacts. Moreover, if edible NBS are to contribute fully to the realisation of productive landscapes in cities that are sustainable and desirable, model-based tools, such as those analysed in this paper, need to be linked back to design and resource aspects. For example, developing tools that make explicit connections between quantifiable outputs, e.g., assessing flooding mitigation of a single site, the spatial role that the site plays in creating a continuous productive urban landscape, and the visual qualities associated with nature-based place making in an urban setting.

Furthermore, the difficulties in reusing models and equations from other tools caused, in the last term, a waste of time and public money. The European Commission is advocating for FAIR principles that pursue proper data sharing. In response to this, the number of publications that include a dataset published in an open repository is increasing. Open datasets are very welcome, but most data related to environmental sciences are context-specific; thus, the datasets are often only reusable in the same context where they were collected. Therefore, we support a step forward, sharing models and equations used to estimate data along with datasets. Most models are not context-specific and can be reused in any context, provided they are fed with the proper data. Then, when they are, they offer at least a basis for creating a new model adapted to the study's context. Although there are some interesting initiatives in this direction, such as the ActonNBS catalogue, the models and equations used by the catalogued tools are usually not available. In summary, public agencies must continue to encourage data sharing but models behind the tools should be open as well, since they are, at least, as useful as the datasets.

At the end of this review, where are we? A single all-encompassing model-based tool has not been identified, but the number of tools currently available indicates a desire to consolidate knowledge collectively and as far as possible accessibly. As the field moves forward, new open access models for edible NBS tools will enable a more informed

and complex dialogue about the value of their outputs, thereby going beyond the users' perceived needs of the tool. As this dialogue unfolds, we can expect to see more diverse edible NBS models, before finding a convergence at agreed quantifiable and qualitative parameters for the evaluation of edible NBS.

**Supplementary Materials:** The following are available online at https://www.mdpi.com/article/10.3390/w13172366/s1, Table S1: Short description of the tools, Table S2: Data items managed by the tools (output data, input and external datasets), Table S3: Tools review database.

**Author Contributions:** Conceptualization, E.M., J.P.-R., M.Š., J.A.C.C., A.V., D.I., N.A., K.B., J.C.; methodology, E.M., J.P.-R., M.Š., J.A.C.C., A.V., D.I., N.A., K.B., J.C.; formal analysis, E.M., J.P.-R., M.Š.; investigation, E.M., J.P.-R., M.Š., A.V., D.I., N.A., K.B., J.C.; resources, E.M., N.A., J.C.; data curation, E.M.; writing—original draft preparation, E.M., J.P.-R., M.Š., J.A.C.C., A.V., D.I., N.A., K.B.; writing—review and editing, E.M., J.P.-R., M.Š., J.A.C.C., A.V., D.I., N.A., K.B., J.C.; visualization, E.M., J.P.-R., J.A.C.C., A.V., K.B.; supervision, N.A., J.C.; project administration, N.A., J.C.; funding acquisition, N.A., K.B., J.C. All authors have read and agreed to the published version of the manuscript.

**Funding:** This research and the APC were funded by EU Horizon Programmes, EdiCitNet project grant number 776665 and the COST Action CA17133 Circular City ("Implementing nature based solutions for creating a resourceful circular city", http://www.circular-city.eu.

**Data Availability Statement:** The underlying data of this article are provided as Supplementary Material.

**Acknowledgments:** We want to acknowledge the projects and researchers that answered our questions about the tools they developed or were developing.

**Conflicts of Interest:** The authors declare no conflict of interest. The funders had no role in the design of the study; in the collection, analyses, or interpretation of data; in the writing of the manuscript, or in the decision to publish the results.

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
