# Peer review of "Tools for Edible Cities: A Review of Tools for Planning and Assessing Edible Nature-Based Solutions"

_water, doi:10.3390/w13172366_

Round 1

Reviewer 1 Report

The paper is much improved and more focused. 

Reviewer 2 Report

The review paper “Tools for edible cities: A review of tools for planning and assessing edible nature-based solutions” has been improved. Thank you for addressing all my concerns.

Line 1: it should be "review" not "article".

This manuscript is a resubmission of an earlier submission. The following is a list of the peer review reports and author responses from that submission.

Round 1

Reviewer 1 Report

In general, a well-written and cogent paper.  It does what it says it will do, and that is not trivial.

Change:

 contrary to Open Science principles stood up by many research public agencies 

to

  contrary to Open Science principles stood up for by many research public agencies 

Elaborate the "new generation of indicators" piece: 

In ad-146 dition, a new generation of indicators is emerging which are also looking at institutional 147 and governance aspects [24]. 148 

Change:

(Framing Concrete and FarmAR app) can support local practitioners in the daily manage-367 

to read:

   Farming Concrete....

Elaborate the idea of "a potential for future research.":

   a significant potential for future research and practice aimed at break-402 ing down disciplinary barriers to enable the lasting implementation of eNBS 

Change:

   as it requires strictly cross-disciplinary approach and as such represents a gap towards 417 

to read:

   requires a strictly cross-disciplinary approach

Elaborate on the idea of "need to be linked back to design and...":

   model-500 based tools, such as those analysed in this paper need to be linked back to design and 501 resources aspects. 

Suggestion:

It would be great to see a nod, at the very least, to a needs-based gap assessment of tools, models and indicators...what do users look for, or need, as compared to what the tools deliver.  To what extent are they satisfactory to the end users?

Reviewer 2 Report

I was disappointed with this paper. I expected an overview of urban agriculture [in the broadest sense] but there are very few takeaways in the current manuscript, and community representatives would be hard pressed to find any information to support their work. I know this is a scholarly journal, but OA should also imply content with broad application.  

I think that there are three fundamental problems with this submission. The first is: what does this have to do with water? It is mentioned briefly in lines 411-414 but that is close to the end of the paper. Logically, this means the paper needs a ‘water focus’ or it should be forwarded to another journal, such, as Sustainability or Foods.

The second problem is: what is this paper actually about? The title points to food and urban agriculture, or ‘the productive urban landscape’ line 25].  But this is never a fixed target. At some junctures, the ‘urban food question’ yields to ‘enhanced climate resilience’ [80], ‘sustainable urban agriculture’ [406], or broader topics such as social justice [144] . Clearly these can be easily linked—but only in a sloppy manner. We need specificity. For instance, while local foods are generally seen to be ‘a good thing’ they tend to use more inputs than factory farms. This might include more water, for example. Using open space for urban agriculture can be linked to green gentrification. And there are open questions about nutritional outcomes [see e.g. the work by Kato on New Orleans, on food sovereignty].   

And the third is: what is the punchline? Now that I have worked through the manuscript, I have no idea what I have learned. I could not use this is an teaching Module or pass it on to sustainability scholar working in a transdisciplinary setting [see below].  

I have some specific concerns. First, I hate the concept of “Edible cities”. It makes as much sense as an edible opera house or an edible bus. Recognizing that the authors were not the originators of this silliness, I encourage something like Celik’s ‘edible landscape’, but given my earlier comments, a broader title such as “Productive urban landscape” [25] connects better to the content. At the very least, ‘edible’ is bad direction to pursue: meat is edible but not at all sustainable, for example. Similarly, why would we be offered KPIs for resilience but not nutrition?

Second: what is the urban food question? [66]. Did we get an answer?

Third: eNBS is misleading, it sounds like it might have a connect to e-commerce [68];

And fourth, some aspects of the discussion need work. No paper should contain the phrase “To the best of our knowledge” [94]: you are supposed to be the experts! Some acronyms appear without explanation (e.g. LCA) [114]. And while the use of language is generally very good, there are some slips [‘stood up by many research public agencies’ [Abstract]; bellow 334; ‘we stand up for a step forward’ [509].

All told, this paper has potential if it would 1. Decide explicitly what it is about; and 2. Do a useable content analysis of the materials that were discovered, following the flow chart in Figure 1. The very general summaries [e.g. figures 3-5] need detailed analysis. The material in Figure 6 is much too cryptic, e.g. I still don’t know why we have two classes of edible NBS, one with 4 entries and one with 16.The material destined for the Appendix should be featured in the paper and given a proper content analysis: this should be the heart of the paper. 

Reviewer 3 Report

Mino et al. examined 32 NBS projects, identifying and characterizing 50 tools. Then, in terms of their format and knowledge domains, the six tools currently accessible and presenting indicators were examined further. Their key conclusion shows that there is a lack of tools available to assist users in designing and implementing edible NBS. The paper is interesting but should be processed as a “review”. Furthermore, I have several suggestions that could improve the manuscript:

Introduction

Page 2 L51-63: You should consider more recent publications on this topic, e.g.

Artmann & Sartison (2018) The Role of Urban Agriculture as a Nature-Based Solution: A Review for Developing a Systemic Assessment Framework Sustainability  10(6), 1937; https://doi.org/10.3390/su10061937

Gottero (2019) Agrourbanism: Tools for Governance and Planning of Agrarian Landscape ISBN

978-3-319-95575-9 https://doi.org/10.1007/978-3-319-95576-6

Russo& Cirella (2019) Edible urbanism 5.0. Palgrave Communications volume 5, Article number: 163 https://doi.org/10.1057/s41599-019-0377-8

Sartison & Artmann (2020) Edible cities – An innovative nature-based solution for urban sustainability transformation? An explorative study of urban food production in German cities, Urban Forestry & Urban Greening https://doi.org/10.1016/j.ufug.2020.126604.

Zeunert & Waterman (2020) Routledge Handbook of Landscape and Food ISBN 9780367502126

Methods: it is not clear how you have identified these tools. Have you used specialist databases to conduct your search?

Figure 6. it is illegible, please increase font size.

Discussion: please discuss tool accuracy and limitations.